# Colonization and Biodegradation Potential of Fungal Communities on Immersed Polystyrene vs. Biodegradable Plastics: A Time Series Study in a Marina Environment

**DOI:** 10.3390/jof10060428

**Published:** 2024-06-18

**Authors:** Aurélie Philippe, Marie Salaun, Maxence Quemener, Cyril Noël, Kévin Tallec, Camille Lacroix, Emmanuel Coton, Gaëtan Burgaud

**Affiliations:** 1Univ Brest, INRAE, Laboratoire Universitaire de Biodiversité et Écologie Microbienne, F-29280 Plouzané, France; aurelie.philippe@univ-brest.fr (A.P.); marie.salaun.lochou@gmail.com (M.S.); maxence.quemener@univ-brest.fr (M.Q.); emmanuel.coton@univ-brest.fr (E.C.); 2Ifremer, IRSI, SeBiMER Service de Bioinformatique de l’Ifremer, F-29280 Plouzané, France; cyril.noel@ifremer.fr; 3CEDRE Centre de Documentation, de Recherche et d’Expérimentations sur les Pollutions Accidentelles des Eaux, 715 Rue Alain Colas, CS 41836, CEDEX 2, 29218 Brest, France; kevin.tallec@cedre.fr (K.T.); camille.lacroix@cedre.fr (C.L.); 4Institut Universitaire de France, France

**Keywords:** fungal plastisphere, foamed polystyrene, diversity, colonization dynamics, bioremediation potential

## Abstract

Plastic pollution of the ocean is a major environmental threat. In this context, a better understanding of the microorganisms able to colonize and potentially degrade these pollutants is of interest. This study explores the colonization and biodegradation potential of fungal communities on foamed polystyrene and alternatives biodegradable plastics immersed in a marina environment over time, using the Brest marina (France) as a model site. The methodology involved a combination of high-throughput 18S rRNA gene amplicon sequencing to investigate fungal taxa associated with plastics compared to the surrounding seawater, and a culture-dependent approach to isolate environmentally relevant fungi to further assess their capabilities to utilize polymers as carbon sources. Metabarcoding results highlighted the significant diversity of fungal communities associated with both foamed polystyrene and biodegradable plastics, revealing a dynamic colonization process influenced by the type of polymer and immersion time. Notably, the research suggests a potential for certain fungal species to utilize polymers as a carbon source, emphasizing the need for further exploration of fungal biodegradation potential and mechanisms.

## 1. Introduction

Global plastic production has exhibited an exponential pattern since 1950, now reaching ~391 million tons (Mt) per year [1]. Plastics are versatile materials, prized for their numerous attributes, including resistance, durability, malleability, and cost-effectiveness [2]. However, their extensive utilization results in the generation of substantial quantities of unmanaged waste. Out of the 8300 Mt of plastic waste generated between the years 1950 and 2015, ~9% underwent recycling, while ~12% were incinerated, and the remaining ~79% were deposited in landfills or natural environments [3].

Oceans represent the final recipients of mismanaged plastic waste with an annual flux estimated at 9–14 Mt per year [4]. Thermoplastic materials, including polyethylene (PE), polypropylene (PP), polyvinyl chloride (PVC), polystyrene (PS), nylon, and polyethylene terephthalate (PET), collectively account for ~90% of the global plastics produced [5], and thus are those that are mostly encountered as debris in the marine environment [6]. PS accounts for ~6% of Europe’s plastic production but a substantial >60% in terms of volume [1]. Among the different types of PS, expanded polystyrene (EPS) and extruded polystyrene (XPS), which both constitute foamed materials with several qualities (e.g., high mechanical compressive strength, insulation properties) consequently find extensive usage in various sectors, ranging from packaging and leisure to aquaculture/fishing and construction [7]. Between 2018 and 2020, in OSPAR countries actively monitoring foamed polystyrenes, including Denmark, the Netherlands, Germany, France, Ireland, and Portugal, EPS and XPS pollution constituted 15% of the total number of plastics and 13% of the overall litter found on beaches [8], raising significant environmental concern. 

Over the last decade, microbial communities inhabiting marine plastic debris (MPD), commonly referred to as the ‘Plastisphere’, have garnered significant scientific attention. These communities are characterized by their distinct diversity and unique composition when compared to those residing in the surrounding seawater. Microbial biofilms associated with MPD are believed to host an estimated 1000 to 15,000 tons of microorganisms [9], a number that may well be conservative given recent estimates of oceanic microplastic counts in the upper ocean that have been reassessed from 5 trillion pieces [10] to 24.4 trillion pieces [11]. Numerous studies have targeted microorganisms from all three domains of life that are linked to MPD, aiming to understand their colonization potential. These investigations have revealed a rapid, well-organized, and dynamic colonization process when plastic materials are immersed in marine environments [12,13,14,15,16]. The majority of studies have concentrated on the bacterial component, with a relatively limited exploration of microeukaryotic communities, particularly fungal ones [17,18,19,20,21,22,23,24,25,26,27,28]. From these investigations, relatively convergent patterns were observed with (i) the geographical locations and inherent environmental factors exerting a more substantial influence on fungal colonization than the polymeric composition of plastic material, (ii) the polymeric composition within a specific type of plastic (conventional or biodegradable) that does not significantly affect richness or community composition, except in rare cases [29], and (iii) the significant impact of immersion duration on fungal colonization, except in rare cases [27], with one specific study highlighting a significant switch in the fungal community structure after 30–40 days of immersion [26]. In terms of taxonomic diversity, Ascomycota predominates in plastic-associated fungal communities. Additionally, various other fungal phyla, including Basidiomycota, Chytridiomycota, Glomeromycota, Mucoromycota, Zoopagomycota, and Cryptomycota, have also been detected [17,23,24,26,27,30], a taxonomic distribution that aligns with the general pattern of marine fungal diversity [31]. To date, although some studies have conducted works on PS collected from the sea [18,23,24], or immersed and collected at a single time point [20,21,22,28], a notable knowledge gap remains as there has been no investigation conducted on the immersion and collection of PS at multiple time points. 

Although more than 400 microbial species have been identified as capable of degrading plastic polymers, evidence of their ability to break down commonly used polymers, such as PP, PS, and PVC, must be viewed with caution [32], most studies failing to differentiate whether the loss of material results from substance leakage such as additives and fillers or effective degradation of the polymers’ building blocks. Among the species documented for their plastic degradation potential, a large majority, accounting for 65.6% of all identified species, are bacteria, encompassing 286 distinct species. These bacteria are categorized within five phyla, specifically Proteobacteria (30.4%), Actinobacteria (20.3%), Bacillota (13.8%), Bacteroidota (0.69%), and Cyanobacteria (0.46%). Regarding fungal organisms capable of plastic degradation, 150 species, constituting 34.4% of the total identified species, are distributed across three fungal phyla: Ascomycota (27.0%), Basidiomycota (4.4%), and Mucoromycota (3.0%). Despite the significance of polystyrene (PS) in the global production of plastics, there appears to be a dearth of organisms identified thus far with the capability to degrade this high molecular weight polymer. PS has been reported to be subject to degradation by 14 bacterial species, primarily belonging to the Bacillales order, and one fungal species (*Talaromyces variabilis*) [33]. Recently, various fungal species, such as *Aspergillus niger*, *Aspergillus flavus*, *Trichoderma* sp., *Aureobasidium* sp., *Penicillium* sp., and *Curvularia* sp. [34], together with *Mucor* sp., *Cephalosporium* sp., *Mortierella* sp., *Geomyces* sp., and *Phaenerochaete chysosporium* [35] were shown to exhibit PS degradation capabilities through weight loss, FTIR spectroscopy, and microscopic examinations. It should be noted, however, that their degradation capabilities, when indicated, appeared low, usually <5%, and that none of the listed putative PS-degraders were isolated from the marine environment. More recently, the white rot fungus *Pyrrhoderma noxium* was demonstrated to cause a 16% reduction in EPS weight after 30 days of incubation [36], highlighting the potential of fungi in degrading PS polymers within a relatively limited timeframe.

These findings illuminate emerging trends regarding distribution and biodegradation patterns of plastic-associated fungi, raising first insights into the comprehensive understanding of fungal colonization on plastics along with their real biodegradation potential. Here, we aimed to address a knowledge gap, particularly regarding the colonization potential and dynamics of fungal communities on immersed PS-type plastics over a time series in the marina of Brest (Brittany, France), in comparison to other plastics, specifically biodegradable alternatives. In this study, six different polymers, i.e., three conventional foamed PS and three biodegradable alternatives, were collected after three, six, and nine weeks of immersion in a marina environment, and associated fungal communities were analyzed using (i) high-throughput 18S rRNA gene amplicon sequencing, to investigate taxa associated to plastics and to the surrounding seawater, and the influence of the polymeric nature on the composition and structure of the fungal communities, and (ii) culturing and screening approaches to isolate environmentally relevant fungi and assess their abilities to utilize different kind of polymers as carbon source. The first aim of the present study was to elucidate differences in fungal colonization between polymers of varying compositions and characterized by different parameters, such as toxicity on marine organisms and the transfer capability of hazardous chemicals as leachates. Additionally, this study aims to determine whether fungal community assembly on marine plastic debris is governed by deterministic and/or stochastic processes.

## 2. Materials and Methods

### 2.1. Immersion Experiments 

A field experiment was conducted in the Marina du Château in Brest (France) (48°22′49.6″ N, 4°29′21.6″ W) from June to August 2021. Pieces of six different plastic materials were immersed in stainless cages in triplicate and exposed to the natural aquatic environment. The experiment was performed over a time series of 9 weeks with 3 sampling time points (after three, six, and nine weeks of immersion). Plastic materials corresponded to three foamed PS, including 2 expanded PS (PSE-CP and PSE-PI) and one extruded one (PSX-BA), as well as three biodegradable alternatives, namely Polylactic acid (PLA), Polylactic Acid + Polybutylene Adipate terephthalate (PLA/PBAT), and Poly3-hydroxybutyrate-co-3-hydroxyhexanoate (PHBH). Each type of plastic is defined by distinct characteristics, including: (i) biodegradability, (ii) ability to absorb organic pollutants, such as PAHs (Polycyclic Aromatic Hydrocarbons) and PCBs (Polychlorinated Biphenyls), and (iii) the extent of its toxic impact on marine life, specifically in terms of cytological, genotoxic, and reprotoxic effects based on a comparative approach among materials [37] (Appendix A). The comparative approach used in [37], called the Marine Impact Assessment Toolkit v1.0, allowed the calculation of scores in the two categories “Transfer of hazardous chemicals” and “Toxicity on aquatic organisms” to compare the 6 materials in the present study (scores are indicated in Appendix A). All plastic materials were cut to similar dimensions (4.25 × 3 cm) and thickness (0.65 cm). Additionally, water samples were retrieved at each time point for (i) temperature, salinity, and pH determination, and (ii) comparison of the fungal communities on the plastic substrates with the natural communities in the surrounding seawater. Water samples (1 L) were filtered on sterile 0.22 μm cellulose nitrate filters (Sartorius Stedim Biotech GmbH, Göttingen, Germany). A total of 54 plastic pieces were collected along with 3 surrounding seawater samples (1 per sampling time point). After sampling, plastics were rinsed with sterile station water and stored at 4 °C in sterile Falcon tubes before processing in less than 3 h. Each plastic piece was then divided into two parts under a laminar flow hood; one half was stored at −80 °C until further processing for DNA extraction while the other half was immersed in 10 mL of sterile station water and stored at 4 °C for culture analysis.

### 2.2. Culture-Independent Approach

#### 2.2.1. DNA Extraction, Amplification, and Sequencing

Sterile DNA extraction was performed from the 54 sampled plastic materials (1 cm^2^ piece, extracted using a sterile scalpel blade taking care to only collect the top 2 mm of the surface) and 3 surrounding seawater samples (entire 0.22 μm filter). DNA extraction was carried out using the FastDNA Spin Kit for Soil (MP Biomedicals, Illkirch, France), following the manufacturer’s instructions. Concentration of extracted DNA was measured with a NanoDrop 1000 Spectrophotometer (Thermo Fisher Scientific, Montigny-le-Bretonneux, France). Two negative controls (reaction mixture without DNA) were included to ensure that no contamination with exogenous amplifiable DNA occurred during the different stages of sample treatment. Semi-nested PCR amplifications of the V4 region of the 18S rRNA gene were performed using the obtained DNA extracts as templates. The nu-ssu-0817 (5′-TTAGCATGGAATAATRRAATAGGA-3′) and nu-ssu-1536 (5′-ATTGCAATGCYCTATCCCCA-3′) primers were used for the first PCR while the nu-ssu-0817/nu-ssu-1196 (5′-TCTGGACCTGGTGAGTTTCC-3′) primer couple was used for the second amplification as described by [26]). PCR products were visualized by agarose gel electrophoresis and then iTAG sequenced at Genome Quebec using Illumina MiSeq PE300 (Montréal, QC, Canada).

#### 2.2.2. Bioinformatics and Statistical Analyses

Raw data were analyzed using the SAMBA v4.0.0, a standardized and automated metabarcoding analysis workflow (https://gitlab.ifremer.fr/bioinfo/workflows/samba, accessed on 19 December 2023) [38]. The workflow comprised three main components: data integrity checking, bioinformatics processes, and statistical analyses. QIIME 2 version 2022.11.1 [39] and DADA2 version 1.26.0 (Divisive Amplicon Denoising Algorithm) [40] were used with parameters optimized by FIGARO version 1.1.2 [41]. Initial processing involved raw data filtering, including primer removal and elimination of reads with incomplete or incorrect primer sequences, using Cutadapt V4.6. [42]. The Amplicon Sequence Variants (ASV) clustering method was applied through DADA2, involving quality filtering, sequencing error correction, pairwise read merging, inference of sample ASVs, and chimera identification. 

The primary objective of this initial stage was to assess data quality by optimizing parameters for trimming reads in microbiome analysis to enhance information extraction from post-trimming sequences, utilizing the FIGARO tool V1.1.2 [41]. To mitigate potential diversity overestimation inherent in DADA2, an additional step of ASV clustering was performed using dbOTU3 V1.5.3 [43]. Subsequently, the resulting ASVs underwent taxonomic assignment through a Naive Bayesian approach against the PR2 V5.0.1. database [44]. In alignment with the approach recommended by [45], the DESeq2 method was chosen as an alternative to rarefaction for data normalization. Moreover, the Analysis of Composition of Microbiomes (ANCOM-BC) statistical framework was employed to compare microbiome compositions. The SAMBA workflow generated an R phyloseq object, used for all diversity analyses (R Version 4.3.2). To assess intra-specific diversity, four diversity indices (Chao1, Shannon, Simpson’s inverse, and Pielou) were calculated. For examining inter-specific differences among samples, beta diversity analysis was conducted using ordination method (Principal Coordinate Analysis—PCoA).

### 2.3. Culture-Dependent Approach

Four distinct media, namely Martin Rose Agar (M), Sabouraud Dextrose (SAB), Potato Dextrose Agar (PDA), and Czapeck (CZP), were employed for filamentous fungi and yeast isolation. These media were formulated with and without 10 g·L^−1^ of six different plastic powders (PS, PE, PET, PVC, PCL, and PHBV), and with or without antibiotics (chloramphenicol and streptomycin at 20 mg·mL^−1^, respectively). The media supplemented with plastic powder contain no other carbon sources except agarose. The polymers supplemented in the culture media differ from those immersed in the marina of the port of Brest. This distinction ensures the reproducibility of results, as the supplemented polymers are sourced from standardized international suppliers. The four conventional polymers supplemented corresponded to polystyrene beads (900 µm size, Goodfellow Cambridge Limited, Huntingdon, UK, ST31-PD-000103), low-density polyethylene powder (300 µm size, Goodfellow, ET31-PD-000131), polyethylene terephthalate powder (300 µm size, semi-crystalline and copolymer, crystallinity > 40%, Goodfellow, ES306031), and unplasticized polyvinyl chloride flakes (250 µm size, Goodfellow, CV316010). The two biodegradable polymers corresponded to polycaprolactone flakes (Powdered Polysciences, Inc., Warrington, UK, 25090-100) and polyhydroxybutyrate/polyhydroxyvalerate 1% biopolymer powder (300 µm size, 490 kg/mol MW, Goodfellow, BV34-PD-000130). Before being supplemented in the different culture media, plastic polymers underwent disinfection in a 70% ethanol bath for 4 h with agitation. Subsequently, ethanol was evaporated using a water bath at 80 °C for 20 min for all plastic polymers, except for PCL, which was treated at 45 °C to prevent the aggregation of plastic particles. Incubation was conducted at 15 and 25 °C, representing the average optimal temperature range for a wide array of fungal species. Overall, this experimental design included 112 different conditions per sample, resulting in a total of 6384 Petri dishes in order to capture the widest possible representation of in situ fungi.

A cell-detachment procedure involving a 3 × 5 s sonication step at 40 kHz using a sterilized probe was performed on collected plastic pieces (3 cm^2^), as described by [46], to detach cells from the plastic matrix. Cell suspensions served as inoculum on Petri dishes (10 μL/Petri dish) and were spread with sterile rakes before incubation. From each Petri dish showing fungal growth, pure cultures were obtained by streaking (for yeasts) and central picking (for filamentous fungi) with similar culture conditions.

### 2.4. Screening for Use of Plastics as a Source of Carbon

#### 2.4.1. Stratified and Weighted Random Sampling 

Given that our culturing approach yielded more than 2000 fungal isolates, we implemented a stratified and weighted random sample strategy to select 300 isolates representative of the initial collection. Within each sampling time point, 100 isolates were subsampled based on their observed abundance in the original media supplemented with the 6 different disinfected polymers (see Section 2.3) and the polymer-free control. To ensure a comprehensive representation, ten fungi growing without plastic in the medium were arbitrarily included among the 100 isolates. Subsequently, 90 isolates were randomly and proportionally selected from the compiled list using the *sample* function of the R studio software v.2023.09.0+463. All isolates were cultured on Bushnell-Haas (BH), a minimum medium, composed of magnesium sulfate (0.2 g·L^−1^), calcium chloride (0.02 g·L^−1^), potassium mono- and di-phosphate (1 g·L^−1^, respectively), ammonium nitrate (1 g·L^−1^), and iron chloride (0.05 g·L^−1^). BH medium was supplemented with 15 g·L^−1^ of artificial sea salts (Sodium chloride NaCl 26.29 g·L^−1^, potassium chloride KCl 0.74 g·L^−1^, calcium chloride CaCl_2_ 0.99 g·L^−1^, magnesium sulfate hexahydrate MgCl_2_-6H_2_O 6.09 g·L^−1^, and magnesium sulfate MgSO_4_-7H_2_O 3.94 g·L^−1^). 

Each of the 300 fungal isolates were first cultured on PDA solid media supplemented with artificial sea salts to check purity of each isolate. Once sufficient growth and spore production were achieved, various strategies were employed to obtain an inoculum, depending on whether the isolates were filamentous fungi or yeasts. For filamentous fungi, spores were obtained by extracting a plug from the thallus using a sterile 6 mm diameter hole punch. The biological material was then carefully separated from the agar with a sterile scalpel and suspended in 5 mL of Bushnell-Haas (BH, see composition above), as a minimal medium. Vigorous agitation (Rotolab, maximum speed, 10 s) was applied to maximize spore detachment from the mycelium. The spore suspension was then diluted to 1/10th to achieve a concentration of approximately 10^6^ spores/mL. As for yeasts, cells were collected by scraping the agar plate with a sterile disposable loop, which was filled to a quarter with yeast cells. The content was also suspended and mixed in 5 mL of BH medium, followed by vigorous agitation. Subsequently, yeast suspensions were also diluted to 1/10th to obtain a similar concentration of approximately 10^6^ yeast cells/mL. 

Two approaches were used in the frame of this screening step aiming to assess the ability of each isolate to utilize plastics as carbon sources: a solid-based approach using agar plates, and a liquid-based approach using a combination of laser nephelometry (NEPHELOstar Plus, BMG Labtech, Ortenberg, Germany) and oCelloScope (BioScience Solutions, Thyborøn, Denmark).

#### 2.4.2. Solid-Based Screening 

BH agar plates were supplemented with 20 g·L^−1^ of agar and the targeted disinfected plastic polymer (see Section 2.3) as carbon source at a final concentration of 10 g·L^−1^. To ensure the validity of our methodology, two controls were included, namely a positive control supplemented with glucose (Sigma-Aldrich, Saint-Quentin Fallavier, France, 1.5% final concentration) and a negative one deprived of any carbon source (except the agar). The negative control was used to highlight the capacity of isolates to utilize agar as a carbon source and/or potential metabolic reserves, and/or to assess necromass utilization.

Each Petri dish was inoculated using the three-point method using 10 µL suspension of spores or yeast cells at each point source inoculation. Each Petri dish was incubated at 25 °C for 14 days, and growth was observed by capturing high-quality images every two days. Images were then analyzed using ImageJ version 1.53 [47] for standardization and extraction of the area of each thallus (3 thalli per condition). An analysis was conducted to identify the most promising isolates using only data after 14 days of growth. To identify the most promising isolates, a highly stringent approach was implemented. Initially, a threshold was set to include isolates with relatively substantial mean areas (set arbitrarily at 200 mm^2^, representing ~20% of the Petri dish or the maximal growth area in a 3-point inoculation) to exclude fungi with limited thallus size. Mean areas obtained from each minimal medium were then compared to those from each condition (BH + either PE, PS, PET, PHBV, PCL, or PVC) to select for isolates demonstrating superior growth capacity on polymer-supplemented media compared to minimal medium containing only agar. The selection of the most promising isolates from the solid-based approach was based on two criteria: (i) the highest mean areas and (ii) the highest ratio obtained by dividing the mean area for a specific polymer by the mean area of the positive control (i.e., with glucose).

#### 2.4.3. Liquid-Based Screening

Laser nephelometry was used as a mid/high-throughput device to evaluate fungal growth parameters by measuring light scattered by particles (unicellular and/or filamentous cells) in 96-wells microplates. For this approach, the BH medium was filtered using a 0.22 µm diameter nitrocellulose filter to eliminate potential bias resulting from the presence of particles. To determine an appropriate plastic concentration without prematurely saturating the signal, an optimization step identified the concentrations leading to a nephelometric signal of ~300 to 500 kRNU (Relative Nephelometric Unit): 0.039% for PHBV, 0.15% for PET, 0.31% for PE and PVC, 0.62% for PCL, and 2.5% for PS. Disinfected polymers (see Section 2.3) were diluted with sterile distilled water (10 g/L) and dispensed into 96-well microplates. Plastic particles were then immobilized at the bottom of the wells through the incubation of the microplates at 85 °C for 4 h to create a thin layer at the bottom of each well. Once evaporation was complete, each well was filled with 180 µL of BH medium, with or without glucose supplementation. Lastly, to prevent bacterial contamination, the microplates were treated with UV radiation for 40 min. Each well was then inoculated with 20 µL suspension of spore or yeast cells. The microplates were read daily for 14 days, using a laser nephelometer (NEPHELOstar Plus, BMG Labtech, Ortenberg, Germany). This approach was complemented with an oCelloScope (BioSense Solutions ApS, Thyborøn, Denmark), which allows generation of microscopic images of each well. This double approach allowed an in-depth analysis of the growth potential of fungal isolates by combining quantitative information (number of fungal particles in each well using laser nephelometry) and qualitative data (microscopic images of each well using an oCelloScope). Throughout the experiment, the microplates were incubated at 25 °C under a humid atmosphere. Data analysis was performed using the MatLab software Version 9.6.0 with a Gompertz model approach which generated growth curves and determined growth parameters, such as growth rate (µ) and maximum growth (Rmax). To identify the most promising isolates, a highly stringent approach was also implemented. Initially, a threshold was set to include isolates with relatively substantial nephelometric Rmax values (set arbitrarily at 600,000 RNU, knowing that nephelometric values range from 0 to 2 million RNU) to exclude fungi with limited growth. µ values were then compared to those of the minimal medium (without any carbon source) and only those exceeding the negative control were selected. The selection of the most promising isolates from the liquid-based approach was based on three criteria, namely the highest µ and Rmax values for each specific polymer-enriched condition, as well as the highest ratio obtained by dividing the µ value of a specific polymer by the µ value of the positive control.

Finally, the top 6 isolates, i.e., those that provided consistent data between the three approaches (solid-based, laser nephelometry, and oCelloScope) and the highest growth and higher ratio both from the solid-based and liquid-based approaches, were selected. These most promising isolates were preserved in the UBO Culture Collection (Université de Bretagne Occidentale Culture Collection, https://ubocc.bio-aware.com/, accessed on 5 April 2024). All other fungal isolates were preserved as a laboratory collection.

### 2.5. Sanger Sequencing

Several genetic markers were amplified and sequenced for taxonomic assignments of the six most promising fungal isolates, after DNA extraction using the KingFisher Duo Prime (Thermo Fisher Scientific, Montigny-le-Bretonneux, France). Using primer pairs ITS1F/ITS4, ACT512F/ACT783R, Bt2a/Bt2b, NL4/NL1, polymerase chain reactions were used to amplify the rDNA internal transcribed spacer (ITS) region, the actin partial region (act), the partial beta-tubulin (tub2) gene, and the D1–D2 domain of the 28S rRNA gene, respectively (Appendix A). Contigs were generated using Geneious v. 11.0.18+10 (Biomatters Ltd., Auckland, New Zealand). Isolates were identified via nucleotide BLAST queries (https://blast.ncbi.nlm.nih.gov/Blast.cgi, accessed on 31 January 2024). Sequences obtained from isolates of interest can be accessed using accession numbers in the GenBank database (Appendix A).

## 3. Results

### 3.1. Fungal Diversity Detected by Metabarcoding Analysis

The 57 analyzed samples (54 plastic samples + 3 seawater controls) generated 91 fungal ASVs (when including the seawater samples) and 83 ASVs (when excluding the seawater samples). The alpha diversity was measured using the Chao1, Shannon, InvSimpson, and Pielou indices (Appendix A). Overall, seawater exhibited the highest diversity values (10, 1.75, 4.6, and 0.76 for the Chao1, Shannon, InvSimpson, and Pielou indices, respectively). When compared to the diversity values of seawater, the diversity indices of PSE-PI appeared to be ~4 times lower, PHBH and PLA about 2.5 times lower, and PSX-BA, PBAT, and PSE-CP roughly 2 times lower. 

Among all evaluated factors, i.e., ‘type of polymer’ (conventional or biodegradable), ‘polymers’ (the six immersed polymers), ‘immersion time’ (T1, T2, and T3), ‘marine toxicity’, and ‘transfer of hazardous chemicals’ (Appendix A), only the ‘type of polymer’ variable (for the Chao1 and InvSimpson indices) showed a significant difference between free fungal communities in seawater from those related to plastics (Appendix A). Using a Tukey HSD test followed by a Bonferroni adjustment, *p*-values were 0.03 for the Chao1 index between seawater and conventional plastics, and 0.05 and 0.03 for the InvSimpson index between seawater and conventional plastics and between seawater and biodegradable plastics, respectively. Biodegradable plastics exhibited forty-five unique ASVs, whereas conventional plastics had twenty-six unique ASVs, and the surrounding seawater contained seven unique ASVs. Ten ASVs were common to both biodegradable and conventional plastics, two were shared between biodegradable plastics and the seawater and one was shared between conventional plastics and the seawater. 

While diversity indices showed no significant difference among the six polymers for all tested metrics, a notable distinction was observed for the Chao1, Shannon, and InvSimpson values between PSE-PI (Appendix A), which exhibited lower index values. This specific polymer displayed the highest value for marine toxicity (0.6) and the second highest value for the transfer of hazardous products (1) (Appendix A). Regarding marine toxicity, increasing values of this trait corresponded to a decrease in ASV numbers. Indeed, for the marine toxicity levels of 0 (PHBH), 0.1 (PLA and PLA-PBAT), 0.3 (PSE-CP), 0.4 (PSX-BA), and 0.6 (PSE-PI), the corresponding ASV counts were 17, 15 and 13, 11, 10, and 3, respectively. Linear regression was performed between ASV number and marine toxicity values, and revealed a highly significant impact (*p*-value = 0.001411), indicating that the higher the polymer toxicity is, the lower the fungal diversity is. Linear regression analysis was also conducted to assess the relationship between ASV number and transfer of hazardous product values. The resulting *p*-value of 0.06262, although slightly above 0.05, indicates a trend towards significance, suggesting that as the polymer’s ability to leach hazardous compounds increases, fungal diversity decreases. The ‘immersion time’ variable (from T1 to T3) without seawater samples revealed no significant differences at the 5% risk threshold, even if an observable j-curve is evident for the Chao1, Shannon, and InvSimpson indices, while an inverted j-curve is observable for the Pielou diversity index (Appendix A). However, caution is warranted when interpreting these trends, suggesting a potential decrease in diversity followed by a stabilization over the immersion duration. T1 was characterized by 27 ASVs, T2 by 30, and T3 by 22. Among them, seven ASVs were shared between T1 and T2, while only two were shared between T2 and T3, suggesting a shift in fungal communities. 

For beta-diversity, Principal Coordinate Analysis (PCoA) based on Bray–Curtis distances revealed a distinct separation in community composition among various conditions, as evidenced by statistically significant differences for the ‘polymers’ and ‘immersion time’ factors with *p*-values corresponding to 0.008 and 0.01 with seawater samples, and 0.008 and 0.002 without seawater samples, respectively (Figure 1). Fungal communities associated with plastics exhibited variations from those in the surrounding water. Using a Tukey HSD test followed by a Bonferroni adjustment, the *p*-value was near to significance (*p*-value = 0.059) between conventional plastics and seawater (Figure 1A). This representation also underscores the increased dispersion observed in PSE-PI and PLA samples in comparison to the less expanded ellipses of other immersed polymers (PBAT, PHBH, PSE-CP, and PSX-BA). PLA exhibited the highest complexity in terms of community composition, while PSE-PI showed the lowest complexity as revealed. The ‘immersion time’ factor was also identified as significant (*p*-value = 0.002), indicating clear differences in communities based on the duration of immersion (Figure 1B). Interestingly, a progressive reduction in ellipse sizes was observable from T1 (3 weeks of immersion) to T3 (9 weeks of immersion), fungal diversity being more important at T1 compared to both T2 and T3, and T2 exhibiting higher diversity than T3. A pairwise multilevel comparison using the R function *adonis* (vegan version 2.6.4) revealed a significant distinction between T1 and T3 (*p*-value = 0.047), without seawater samples.

Conversely, no statistically significant difference was discerned for the ‘type of polymer’ (conventional vs. biodegradable), ‘marine toxicity’, and ‘transfer of hazardous chemicals’ variables. When the seawater samples were not considered, a pairwise multilevel comparison using *adonis* revealed a significant distinction between PLA and PSE-PI (*p*-value = 0.029), as well as between PLA and PSX-BA (*p*-value = 0.044) and between PLA and PHBH (*p*-value = 0.015), or close to significance between PLA and PSE-CP (*p*-value = 0.055) and between PLA and PBAT (*p*-value = 0.054). 

Bubble plots were employed to illustrate taxonomic diversity at the phylum and genus levels for the ‘polymers’ and ‘type of polymer’ factors (Figure 2). The Ascomycota, Basidiomycota, and Chytridiomycota phyla were prevalent across all conditions, appearing on both conventional and biodegradable plastics (Figure 2A). Globally, Ascomycota accounted for 73.37%, Basidiomycota for 15.05%, Chytridiomycota for 11.87%, and Zoopagomycota for 0.72% of the observed fungal diversity. PBAT and PSX-BA specifically supported ASVs affiliated with Ascomycota and Basidiomycota, while PLA was associated with ASVs affiliated with Ascomycota and Chytridiomycota (Figure 2B). Zoopagomycota appeared exclusively on conventional plastics. Ascomycota was the most abundant phylum across polymer types, except in PLA and seawater, for which Chytridiomycota was the dominant phylum with 6.53% and 2.09% prevalence, respectively. PLA and PSE-PI were similar in that neither harbored Basidiomycota ASVs. Overall, conventional plastics exhibited a greater diversity of phyla (49.96%) compared to biodegradable plastics (46.26%), with seawater showing the least diversity at 3.77%.

Among the top 10 genera, *Cladosporium* appeared as the most abundant on plastics, showing varying abundance depending on the considered sample type, with 11.84% on conventional plastics, 6.17% on biodegradable plastics, and almost absent on surrounding seawater (Figure 2C). The *Didymella* genus was the next most abundant, with 11.18% on biodegradable plastics and 1.98% on conventional ones. Representatives of the order Chytridiales and the genus *Fusarium* showed some preferences, with 6.53% on PLA, and 6.05% on PSE-CP, respectively (Figure 2D).

Immersion time significantly influenced phyla abundance (Figure 3). Relative abundance decreased over time, with T1 representing 46.56%, followed by T2 at 33.90%, and T3 at 19.54%. Over the course of the three immersion periods, the Ascomycota and Basidiomycota represented 73.37% (T1 = 36.72%, T2 = 20.36%, and T3 = 16.28%) and 14.05% of phyla abundance (T1 = 3.40%, T2 = 8.11%, and T3 = 2.54%), respectively. Chytridiomycota was noted at T1 and T2 with 6.44% and 5.43% respectively, while Zoopagomycota was only detected at T3.

Among the genera observed across all three immersion times, only *Cladosporium* and *Cyphellophora* were consistently present. The abundance of *Cladosporium* decreased with longer immersion times, from 12.60% at T1 to 3.31% at T2, and further to 2.10% at T3. At T1 and T2, genera such as *Didymella*, *Malassezia*, and *Zygophlyctis* were present, along with representatives from the Chytridiales order. The *Fusarium* genus was observed at T2 and became more prevalent by T3. Specifically, T3 marked the exclusive presence of *Leptosphaeria* and *Penicillium*, with abundances of 1.48% and 0.24%, respectively.

Finally, when looking at the ‘marine toxicity’ parameter, only representatives of the Ascomycota phylum were present among all marine toxicity values (Appendix A). Notably, representatives of the *Cladosporium* genus were observed on all polymers, except PHBH. Representatives of the other major phyla, i.e., Chytridiomycota and Basidiomycota, were absent on PSX-BA (toxicity value of 0.4) and PSE-PI (toxicity value of 0.6). 

### 3.2. Assessing Fungal Colonization Potential through Cultivation

All processed plastic samples resulted in the isolation of fungi. The number of isolates obtained from conventional and biodegradable polymers was 1446 and 1101, respectively, whereas seawater samples yielded only 23 fungal isolates. Specifically, the PSE-PI polymer isolated 546 isolates (mean = 182, sd = 49, n = 3), PSE-CP samples resulted in 477 isolates (mean = 159, sd = 54, n = 3), and PSX-BA yielded 422 isolates (mean = 141, sd = 41, n = 3). The biodegradable polymers also exhibited isolation potential, generating 458 fungal isolates with PLA (mean = 153, sd = 49, n = 3), 370 with PLA/PBAT (mean = 123, sd = 21, n = 3), and 273 with PHBH (mean = 91, sd = 14, n = 3). The variability in isolation frequencies, depending on all original plastic materials, appeared substantial, resulting in no statistical difference between the isolation frequencies of the six immersed polymers (ANOVA, *p*-value = 0.19). Interestingly, when frequencies of a similar type (conventional or biodegradable) were pooled, the *p*-value decreased to 0.071, indicating an absence of statistical difference stricto sensu, but still a trend, as the *p*-value was between 0.05 and 0.1. However, such a result should be interpreted with caution as these isolation frequencies include duplicates, which may alter the highlighted pattern.

Isolation rates were subsequently compared using a bubble plot-based figure, considering the specific media employed (i.e., Martin Rose Agar (M), Sabouraud Dextrose (SAB), Potato Dextrose Agar (PDA), and Czapeck (CZP)), as well as the six supplemented polymers (i.e., PS, PE, PET, PVC, PCL, and PHBV). An additional condition excluding any polymer (NP) was also incorporated into the analysis (Appendix A). The M and SAB culture media facilitated the isolation of a high number of fungal isolates, constituting ~33% and ~31% of all isolates. Conversely, CZP and PDA enabled the isolation of nearly similar numbers of isolates (~17.8%). Variations in the frequencies of the ‘Supplemented polymers’ variable across the ‘Media’ variable were observed (Appendix A), suggesting specific preferences of fungal isolates for certain culture conditions. Conditions supplemented with PE and PCL exhibited high isolation frequencies, indicating a potential affinity of fungal isolates for these culture conditions or a higher ability to utilize these plastics as carbon sources. Conversely, the conditions supplemented with PS appeared to have relatively low isolation frequencies overall.

When assessing the correlation between immersed polymers and immersion times (T1, T2, and T3), an inverted j-curve trend became apparent across all examined polymers (Appendix A). Specifically, there was an observed increase in the number of isolates from T1 to T2, followed by a decrease from T2 to T3. The increase between T1 and T2, encompassing all polymers, ranged from 1.24 to 1.98 times. Notably, the increase for conventional polymers (mean = 1.53; sd = 0.39) surpassed that of biodegradable polymers (mean = 1.42; sd = 0.07), albeit displaying greater variability. Subsequently, between T2 and T3, the decline was more conspicuous for conventional polymers (mean = 0.74; sd = 0.16) compared to biodegradable polymers (mean = 0.60; sd = 0.31). Chi-square independence tests, integrating all polymers and incubation durations, underscored a significant dependence between these two factors (*p*-value = 6.34 × 10^−10^). When combining isolate proportions by polymer type (conventional vs. biodegradable), the associated *p*-value for the independence test was 0.14, signifying independence between ‘polymer type’ and ‘immersion duration’. These findings indicated the inherent specificity of each polymer, a characteristic masked when proportions are aggregated into categories.

When evaluating the correlation between immersed polymers and supplemented polymers across various culture media, it was evident that the condition without polymer supplementation consistently resulted in the highest number of isolates (Appendix A). The exception was the PSE-CP condition, for which the number of isolates obtained on media supplemented with PE was slightly higher (22.2% vs. 21.6%). Regarding supplemented polymers, the trends remained generally similar, regardless of the initially immersed plastic. Specifically, PCL and PE led to the highest number of isolates, while the PHBV and PS conditions yielded the lowest numbers. Chi-square independence tests were once again conducted, initially incorporating all polymers. This analysis demonstrated the dependence between the two factors, namely the type of immersed polymers and the supplemented polymers (*p*-value = 2.98 × 10^−3^). When grouping proportions into the ‘conventional’ and ‘biodegradable’ categories, independence was observed (*p*-value = 0.44). This further supports the earlier conclusion, particularly emphasizing the inherent specificity of each polymer.

Ultimately, the influence of the incubation temperature (15 or 25 °C) did not reveal any statistically significant differences; the proportions remaining consistently similar.

### 3.3. Screening for the Ability to Utilize Polymers as Carbon Sources

The culture-dependent approach used generated exactly 2570 fungal isolates obtained from the immersed plastic samples, from which 300 were randomly selected to screen their ability to utilize plastic polymers as carbon sources (Figure 4). Considering the nearly equal original proportions across sampling dates (T1, 3 weeks/T2, 6 weeks/T3, 9 weeks), we opted to arbitrarily choose 100 isolates per sampling date, integrating 10 isolates from the ‘plastic-free’ condition. As shown in Figure 4A,B, original proportions were preserved using stratified and weighted sampling to select isolates representative of the initial collection. 

Screening of representative isolates from the initial collection was performed using three complementary approaches (solid-based—i.e., using agar medium—and liquid-based using laser nephelometry and an oCelloScope). A highly stringent approach was performed in order to select the most promising isolates, meaning those for which consistent results were obtained between each approach and presenting the highest growth capabilities in the polymer-supplemented conditions. Our strategy was linked to the presence of isolates that exhibit strong plastic utilization capabilities as a carbon source, both under liquid and solid conditions, as well as demonstrating important growth under contrasting conditions of plastic particle concentration (10% for the solid-based approach and 0.039–2.5% for the liquid-based environments). We acknowledge that while employing such a strategy, we may overlook potentially interesting isolates that stand out in one or two conditions but consider that, due the large number of isolates and conditions, consistency between approaches should be prioritized. 

An UpSet plot was created to represent the six most-promising isolates selected based on the nuanced principles (Figure 5). Isolates C1589, C1591, C2281, and C2559 showed capability of utilizing only one specific polymer as carbon source; PE, PVC, PHBV, and PCL, respectively. Conversely, isolate C1666 showed strong utilization of PCL, PET, and PS as carbon sources, while C2218 demonstrated growth on PHBV and PVC. 

Isolate C1589 exhibited a fungal growth on solid medium supplemented with PE similar to that on the positive control (solid medium supplemented with glucose) (Appendix A). This visual observation was further supported quantitatively by determining the thallus area, which was in fact slightly smaller (28%) compared to the positive control. The liquid-based method confirmed this trend, demonstrating a growth curve for PE that was slightly lower but comparable to that of the positive control (although the µ was ~40% lower for PE compared to the positive control) (Appendix A). Additionally, a distinctive mycelial network was observed using the oCelloScope for the PE-supplemented condition. 

On solid medium supplemented with PVC, isolate C1591 showed fungal growth that was quite similar to that seen on the positive control (Appendix A). The thallus area was 30% lower in comparison to the positive control. Laser nephelometry results showed that the growth curve on PVC was similar to that on the positive control but still lower (µ ~ 56% lower for PVC than the positive control) (Appendix A). On day 14, the PVC-supplemented condition showed growth structures using the oCelloScope, such as spore chains and conidiospores (Appendix A). 

Despite having a larger thallus in the positive control condition, isolate C1666 showed significant fungal growth on media supplemented with PCL, PET, and PS (Appendix A). Compared to the positive control, the mycelium area measurements were higher for the PET-supplemented condition (38% lower to the positive control), PS-supplemented condition (49%), and PCL-supplemented condition (69%), despite relatively moderate growth curves in liquid media (Appendix A). However, on day 14, a mycelial network was discernible with the oCelloScope (Appendix A). On solid medium supplemented with PHBV and PVC, isolate C2218 showed fungal growth that was nearly identical to that seen on the positive control (Appendix A). The thallus area was reduced compared to the positive control (24% and 14%, respectively). Laser nephelometry results supported this pattern, showing a growth curve for PHBV that was greater than that of the positive control (µ was 10% higher than the positive control) and for PVC that was slightly lower than the positive control (µ was 12% lower compared to the positive control) (Appendix A). Results from the oCelloScope analysis also revealed the presence of reproductive structures, such as spore chains (Appendix A). In comparison to the positive control, isolate C2281 demonstrated reduced fungal growth on solid medium supplemented with PHBV, which led to a nearly 40% reduction in thallus area as compared to the positive control (Appendix A). Additionally, a shift in morphology was apparent with aerial pigmented thalli on PHBV-enriched medium compared to white thalli on the positive control. This pattern was supported by analysis in liquid medium, which revealed a poorer growth curve for PHBV in comparison to the positive control (µ was 66% lower for PHBV) (Appendix A). Still, a strong mycelial network could be seen with the oCelloScope (Appendix A).

On solid medium supplemented with PCL, isolate C2559 displayed fungal growth that was similar to that seen on the positive control (Appendix A). Thallus area comparison showed nearly equality (98%) between this condition and the positive control, thus confirming this observation. However, analysis in liquid media showed a more pronounced reduction (Appendix A). Compared to the positive control, PCL had a 33% lower µ on its growth curve but the oCelloScope analysis revealed a distinct mycelial network (Appendix A).

Nucleotide BLAST queries of the six most promising isolates allowed the identification of C1589 as *Penicillium velutinum* (ITS, 99.83% ID; partial 28S, 100% ID; beta-tubulin, 100% ID), C1591 as *Penicillium crustosum* (ITS, 100% ID; partial 28S, 100% ID; beta-tubulin, 100% ID), C1666 as *Gliomastix murorum* (ITS, 99.83% ID; partial 28S, 100% ID), C2218 as *Aspergillus fumigatus* (actin, 100% ID), C2281 as a putative novel species in the *Penicillium* genus (ITS, 99.83% ID with *Penicillium* sp.; partial 28S, 99.82% ID with *Penicillium* sp.; beta-tubulin, 97.77% ID with *Penicillium* sp.; and 95.94% ID with *Penicillium koreense*, as the first listed species in the top-BLASt hits), and C2559 as *Penicillium brevicompactum* (ITS, 100% ID; partial 28S, 100% ID; beta-tubulin, 98.92% ID).

## 4. Discussion

Our study sheds new light on the diversity and ecology of fungi associated with various plastic polymers immersed in the marine environment in natural settings. Key findings include: (i) differentiating polymer and free-living fungal communities, (ii) evaluating the impact of plastic types (conventional vs. biodegradable), (iii) assessing how immersion time affects fungal colonization, and (iv) evaluating the ability of isolated fungi to utilize polymers as a sole carbon source. We reveal that plastics offer a unique ecological niche for surface-associated fungi in marine ecosystems, establishing these fungi as integral components of the plastisphere, and that some could be potential candidates for plastic biodegradation.

### 4.1. The Fungal Plastisphere

A singular genetic marker was employed in this study, warranting a special note. This choice stemmed from technical limitations in sequencing, which precluded the acquisition of ITS data, thus restricting our analysis to 18S data exclusively. Despite the recognized potential of utilizing two markers to unveil greater diversity, as recently underscored [23], both 18S and ITS datasets yielded congruent trends in a previous work of the fungal plastisphere [26]. Moreover, within the spectrum of eleven publications scrutinizing fungal communities linked to marine plastic debris [17,18,20,21,22,23,24,25,26,27,28], nine relied upon the 18S marker. This alignment facilitates a more robust comparison of our findings with the existing literature. Lastly, it is noteworthy that the 18S marker is adept at elucidating basal fungal lineages, a feature recently highlighted [23].

Although a limited number of ASVs were obtained in this study, the obtained 18S rRNA gene metabarcoding dataset revealed a clear difference in fungal diversity and richness between biodegradable and conventional plastics compared to the surrounding seawater. This result appears consistent with previous studies that demonstrated significant differences between eukaryotic/fungal communities colonizing plastic particles and the free-living seawater communities [18,20,21,25]. Polymer samples immersed in the Marina du Château in Brest, either conventional or biodegradable, had a significantly lower diversity than surrounding seawater. This aligns with previous findings, notably in the Warnow River and Baltic Sea ecosystems where fungal communities associated with immersed PE and PS plastics have shown less diversity than that of the surrounding water [20], a trend that was also observed for other eukaryotic organisms on the same microplastics [21]. However, other studies have shown opposite trends, either for bacteria [48,49] or fungi [17,50]. A previous study highlighted that plastic matrix colonization faces greater disruption offshore compared to harbor settings, attributed to the minimal mechanical force effects in sheltered zones [17] and proposed that biofilms on plastics in such areas achieve rapid maturity.. They propose that biofilms on plastics in such areas achieve rapid maturity. Several studies have demonstrated that open sea habitats exhibit significantly higher richness and diversity of bacterial and microeukaryotic communities on plastic surfaces compared to that of the surrounding water [18,34]. Conversely, coastal areas show greater richness and diversity of bacterial and microeukaryotic communities in seawater than on plastic matrices [21,51,52,53,54,55,56].

Differences between polymers were mostly related to PSE-PI, for which plastispheres showed relatively low values in terms of richness and diversity indices. Our study showed a notable reduced ASV number for conventional polymers that exhibit high toxicity and hazardous chemical transfer, compared to biodegradable polymers with higher values for these parameters. Out of the six tested plastic polymers samples, only this plastic, PSE-PI, is not food contact certified [37]. PS is a C-C backbone polymer with diverse impacts on various microorganisms. PS microplastics were found to induce effects ranging from mortality of *Ceriodaphnia dubia* and *Daphnia magna* to impacts on growth and development on *Chlamydomonas reinhardtii* and *Scenedesmus quadricauda* through alteration of photosynthesis [57]. Drawing clear conclusions on this ecotoxicological part from our study is challenging due to the complex influences on diversity in PS-based plastics. These influences include hydrophobic matrix effects, intrinsic chemical properties of PS, such as PS nanoparticles release, and/or the presence of adsorbed toxic compounds.

Observed trends in fungal diversity and community composition over time indicated that the biofilm evolves towards a reduction of its complexity, as inferred from the absence of several fungal lineages affiliated with the Chytridiomycota and Basidiomycota phyla after 9 weeks of immersion. This trend appears consistent with a previous study targeting PE sheets and dolly ropes immersed in the North Sea, and for which a decrease in fungal relative abundance was observed ~9–14 weeks after immersion, albeit no clear temporal profile could be identified due to considerable fungal community structure variability [17]. However, our results contrast with our prior study on PVC-immersed panels [26], which reported no disappearance of specific fungal lineages over time. 

When comparing results from molecular and culture-dependent approaches, knowing that we can only compare general trends here since the 2570 isolates have not been fully identified, a number of consistencies as well as differences appear. More isolates were obtained from conventional plastic samples than from biodegradable plastics, although the differences were not significant, in line with the diversity indices obtained through metabarcoding. Nonetheless, the number of isolates over time did not follow the same trend between the two approaches used, with a j-curve trend for metabarcoding and an inverted j-curve trend for the cultural approach. Again, these conclusions must be tempered since the isolate collection must contain replicates, which may alter the highlighted trends. However, it appears that supplementation with PS led to a very low number of isolates compared to the other supplemented polymers. This could be correlated, with all due caution, with the low complexity of the fungal communities highlighted for conventional polymers in our metabarcoding dataset, notably for PSE-PI. While PS is a polymer that is refractory to hydrolysis, preventing efficient degradation and leading to persistence in the natural environment [35], various fungal species have demonstrated PS degradation. However, efficiency was limited as revealed by mass loss (*w*/*w*) of 1.81% for *Mucor* sp. and 2.17% for *Cephalosporium* sp. [58], 2.2% for *Mortierella* sp., 6.8% for *Geomyces* sp., and 8.4% for *Penicillium* sp. [59], and the highest potential for *Phanerochaete chrysosporium* with 19.71% degradation [60]. This may explain why few isolates were obtained on PS-supplemented media. 

Ascomycota and Basidiomycota were found on conventional and biodegradable plastics at all immersion times, which is in line with research done in the North Sea (ref). On all polymers, Ascomycota were the most prevalent group, with a lesser percentage of Basidiomycota absent from PLA and PSE-PI plastics. According to a recent study [25], these groupsare frequently found colonizing plastic-associated biofilms. Given that fungi are known to colonize plastics early on, it is possible that they play a significant role in the production of biofilms [17]. Ascomycota dominate among fungal isolates, according to recent studies on the microbial degradation of plastic [33,61]. During immersion times T1 and T2, Chytridiomycota were less prevalent. Chytridiomycota have been detected in low abundance in two investigations, despite evidence of its large abundance on plastic biofilms being recorded in the oceans [20,25]. These are saprotrophs that could be important for the decomposition of organic debris in biofilms [62]. As first reported by [23] in the marine plastisphere and then by [27], Zoopagomycota, representing a minor fraction, were found only on PSE-CP plastics at T3, and not in the surrounding seawater. While there was no significant difference across the three immersion times, there was a nearly significant difference between T1 and T3 [27], indicating that these groups were already well established during the first incubation period (30 days) because they did not find any variations in the composition of the fungal community between incubation periods. Additionally, there were no clear temporal changes in the composition of the fungal community in the North Sea between sampling times [17].

Based on the metabarcoding approach results, the most prevalent fungal genera on plastics corresponded to *Fusarium*, *Didymella*, Chytridiales *incertae sedis*, and *Cladosporium*. *Cladosporium*, which is extensively distributed in the marine environment, was observed on plastics but not in seawater, and some of its species have been reported to exhibit polyurethane polyester (PU)-degrading capabilities [23]. The *Fusarium* genus harbors species that are well known for their capacity to degrade plastic [63]. Only at T3 did the *Penicillium* genus appeared; some of its members having also been reported as degraders of different types of plastics [63]. 

### 4.2. Ability to Utilize Polymers as Carbon Sources

Among the six isolates considered as most promising based on our multiparametric selection strategy, i.e., based on the results from growth tests in Petri dishes and laser nephelometry coupled with oCelloScope analysis, some trends were revealed. Indeed, three isolates were obtained from conventional immersed plastics (two on PSE-CP and one PSX-BA) and three from biodegradable plastics (two from PLA and one from PLA/PBAT), predominantly at longer immersion times (6 and 9 weeks), with only one promising isolate obtained after 3 weeks of immersion (i.e., isolate C2281). A form of consistency between the original culture medium and the ability to degrade a specific polymer can be highlighted. Indeed, C2281 and C2218 isolates came from a medium enriched with PHBV and showed abilities to use PHBV as a carbon source, and C1666 came from a medium enriched with PS and was able to grow on PS as a carbon source. Interestingly, all highly promising isolates were obtained from media enriched with specific polymers, with none from non-enriched media. Isolate C2559 was obtained from a medium enriched with PS but was isolated from PLA, a biodegradable plastic. It showed an ability to utilize PCL, another biodegradable plastic, as a carbon source.

The screening approach has revealed lower but real carbon utilization capacities from polymers compared to positive controls, which appears consistent from a metabolic perspective. While the shortlist of promising isolates may appear limited initially due to stringent criteria, this appears consistent with a recent study on PCL degradation, which found five strong PCL-utilizing isolates among 146 screened and 262 isolated fungi [64]. The most promising isolates now require in-depth complementary analyses to validate the biodegradation of polymer(s) through various dedicated methods (e.g., mass weight, FTIR, ^13^C-labelled polymers, etc.) as it was performed on *Alternaria alternata* FB1 [65], *Zalerion maritima* [66], or *Rhodotorula mucilaginosa* [67] on PE.

### 4.3. Taxonomic Identification of Putative Fungal Degraders

Four of the six most promising isolates are affiliated with the genus *Penicillium*, which clusters numerous species well known for their bioremediation potential [68]. As opportunistic saprophytes, representatives of this genus are widespread and play major roles in a variety of environments [69]. *Penicillium* species are capable of producing extracellular enzymes and metabolizing hydrocarbons in both saline and non-saline environments [68]. Many isolates within this genus have also demonstrated the ability to degrade a variety of polymers. *Penicillium brevicompactum* C2559 was isolated from a nine-week-immersed PLA sample, which corresponds to a PLA sample that also had an ASV associated with the genus *Penicillium* in our metabarcoding data, and cultured using on a PS-supplemented medium. A previous study has demonstrated the ability of this species to degrade polyvinyl alcohol (PVA) [70], a polymer considered as biodegradable [71,72], which appears consistent with our results demonstrating a potential for another biodegradable polymer, here PCL [73]. Such capabilities may be explained by their shared properties. *Penicillium velutinum* C1589 was isolated from a six-week-immersed PLA/PBAT sample and cultured using a PET-supplemented medium. *P. velutinum* C1589 appears able to utilize PE as a carbon source. *Penicillium velutinum* is a ubiquitous species, found both in soil and in marine environments associated with wooden substrates [74]. To date, no study has demonstrated its plastic degradation potential. *Penicillium crustosum* C1591 was obtained from a six-week-immersed PSE-CP sample and cultured using a PET-supplemented medium. *Penicillium crustosum* C1591 appears able to utilize PVC as a carbon source. *P. crustosum* is another ubiquitous species, retrieved in numerous habitats, and recently highlighted as able to degrade PET through the synthesis of cutinases and lipases [75]. Despite less pronounced growth on a PET-enriched medium (growth rate below the applied threshold), this isolate was selected for its ability to grow on both plastics, as confirmed by nephelometry and agar culture methods (Appendix A). Finally, *Penicillium* sp. C2281 appears taxonomically original based on the ITS, partial 28S, and beta-tubulin genetic markers. Complementary analyses are required for complete description of this putative new *Penicillium* species, which was cultured on a PHBV-supplemented medium and showed ability to utilize PHBV as a carbon source. *Penicillium* sp. C2281 seems to belong to the Lanata-Divaricata section, a group gathering many species isolated both from terrestrial and aquatic environments [76].

Our research also revealed another species, *Gliomastix murorum* C1666, collected from a PSE-CP sample that was immersed during six weeks and grown on a medium supplemented with PS. Unfortunately, our metabarcoding dataset did not reveal any ASV affiliated with the genus *Gliomastix*, a discrepancy that may be attributed to potential biases such as the low abundance of this taxon on the plastic matrix. Interestingly, *Gliomastix murorum* C1666 appears able to utilize PCL and PET, but also PS as carbon sources, highlighting here the consistency between the original culture medium and the ability to degrade a specific polymer. The genus *Gliomastix* gathers ubiquist species retrieved in both terrestrial and marine habitats [19]. The ability of this genus to biodegrade plastic has not yet been investigated, to the best of our knowledge, even if one laccase, also known to be involved in plastic degradation (e.g., [77]), from this species has been isolated and described but only in the frame of dye decolorization [78].

Among the numerous fungal species capable of degrading plastic polymers, the *Aspergillus* genus, a filamentous fungal group with representatives known for their broad distribution across various environments, including marine ecosystems [79], stands out. *Aspergillus fumigatus*, in particular, has been recognized for its capacity to degrade a wide range of polymers, such as PHB, PHV, PES (polyethersulfone), PEA (polyesteracetals), PBA (polybutanamide), PCL (polycaprolactone), and PBS (polybutylene succinate) [80], as well as PVC [81] and PE [82]. Here, an isolate identified as *A. fumigatus* (C2218), retrieved from a PLA sample submerged for nine weeks, a sample also showing an ASV linked to *Aspergillus* in our metabarcoding data, was cultured on a medium supplemented with PHBV. Our screening process confirmed its capability to utilize PHBV as a carbon source, highlighting the consistency between cultural and screening approaches. Additionally, *A. fumigatus* C2218 demonstrated an ability to use PVC as a carbon source, aligning with existing literature, and showed a limited capacity to metabolize PS, corroborating the detection of an *Aspergillus*-affiliated ASV in one PSE-CP sample from our metabarcoding dataset. *Aspergillus fumigatus* is a ubiquitous opportunistic pathogen, which can induce a spectrum of infections in humans, with symptoms varying based on the host’s immune status [83]. *A. fumigatus* is known for its capacity to form biofilms, characterized by intricate networks of filaments [84], together with the hydrophobic feature of its conidia, allowing stronger adhesion to hydrophobic surfaces [85]. This might clarify why several isolates of *A. fumigatus* have been detected in different landfills, showing capabilities to degrade various polymers like PE, PVC, PCL, and PHB [81]. *A. fumigatus* is found in terrestrial and marine settings, including seawater, sediment, and in association with algae, corals, and clams, with research primarily focusing on its secondary metabolites. However, reports of its occurrence on marine plastics remain extremely limited [86]. Recently, Gkoutselis et al., 2024 [87] indicated a significant selection process for important human pathogens within the soil plastisphere, closely associated with general virulence traits in fungi. These results support the hypothesis that the plastisphere facilitates niche expansion for fungal pathogens, showcasing the development of a preference among these pathogens for plastic environments. The detection of pathogenic fungi on plastics immersed in seawater broadens the scope of the risk associated with plastic pollution. Our results indicate that the dissemination of fungal pathogens is not confined to terrestrial environments but also extends to marine ecosystems. This expansion of our understanding emphasizes the need for comprehensive strategies to mitigate the impact of plastic pollution across all ecosystems and highlights the interconnectedness of terrestrial and marine environmental health issues.

A notable observation is the relative inconsistency observed between the identified taxa and the metabarcoding dataset. While we have established congruence between these two methods for *Aspergillus fumigatus* and *Penicillium brevicompactum*, as evidenced by ASVs associated with these genera being detected on the same polymers, it is crucial to acknowledge disparities for other *Penicillium* species and *Gliomastix murorum*. Various sources of bias are evident, including (i) the challenge of detecting low biomass through metabarcoding, despite the presumed sufficiency of a single spore for growth on culture media, (ii) the complexity of DNA extraction from resistant spores prior to molecular analysis, and (iii) the inherent heterogeneity in fungal distribution on plastic surfaces. Our metabarcoding analyses were conducted on 1 cm^2^ samples (direct DNA extraction), whereas in culture, cells were detached from larger pieces (equivalent to 3 cm^2^). Thus, one plausible explanation for the observed differences in species richness between these methodologies could be the uneven distribution of fungal microorganisms on plastics, leading to less straightforward detection in metabarcoding compared to culture.

## 5. Conclusions

This study provides significant insights into the colonization patterns and biodegradation potential of fungal communities on plastics immersed in the marine environment. The observed diversity among fungal species on both polystyrene and biodegradable polymers, coupled with their ability to utilize these polymers as a carbon source, underscores their capacity to colonize a variety of plastic matrices, and the potential role of fungi in mitigating plastic pollution in marine ecosystems. These findings highlight the need for further research into the mechanisms of fungal biodegradation and their practical applications in environmental management strategies aimed at addressing the global challenge of plastic waste. Furthermore, the study identified certain fungal species capable of degrading plastics. Some fungal species have shown consistency between their biodegradation capabilities and the polymers from which they were isolated, whereas other species have shown contrasted patterns. With all due caution, as only few isolates were identified, our results may suggest that fungal community assembly on marine plastic debris are a mix of deterministic processes (niche theory) and stochastic processes (neutral theory), as recently proven on the fungal soil plastisphere using metagenomic data [87]. Interestingly, no species with ability to utilize polymer as carbon source was isolated from PSE-PI, the polymer exhibiting the highest toxicity and lowest fungal diversity in our metabarcoding dataset. One species with the ability to utilize polymer as carbon source is also known as an opportunistic pathogen (*Aspergillus fumigatus*). This dual role underscores the complexity of fungal interactions within marine ecosystems and highlights the potential risks and benefits of leveraging these organisms in bioremediation efforts. It emphasizes the necessity for careful consideration of ecological impacts and health implications in the development of fungal-based strategies for plastic waste management, which need to be more focused on enzymes than microorganisms.

## Figures and Tables

**Figure 1 jof-10-00428-f001:**
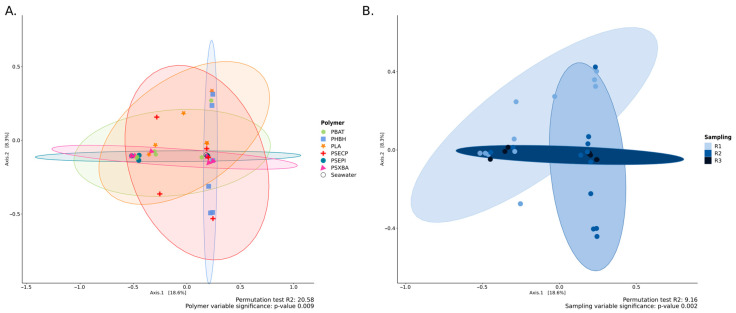
Principal Coordinate Analysis (PCoA) based on the 18S dataset, showing the relationship between the variables “polymers” (**A**) and “immersion time” (**B**). The PCoA vas generated using Bray–Curtis dissimilarity as the distance measure, with each point representing a sample. Ellipses connect groups of samples that were subjected to the same conditions.

**Figure 2 jof-10-00428-f002:**
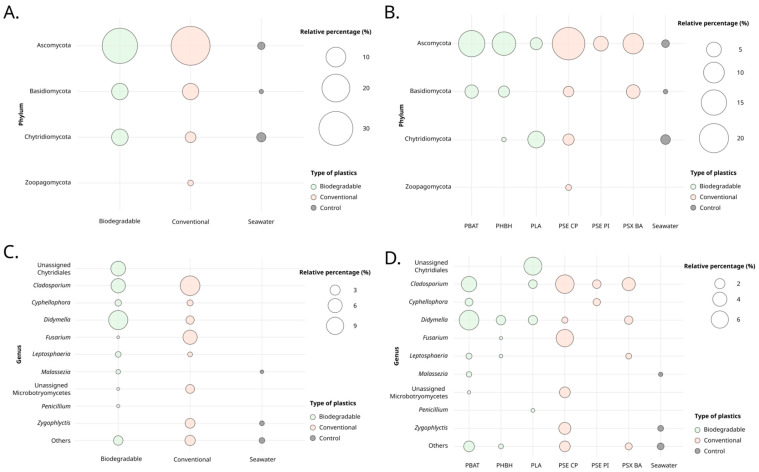
Bubble plot representing the proportions of phyla or genus, based on the 18S dataset, according to the “Polymers” (**A**,**B**) and “Type of plastic” (**C**,**D**) variables.

**Figure 3 jof-10-00428-f003:**
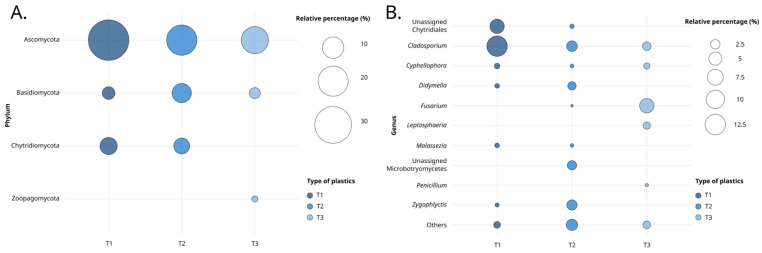
Bubble plot representing the proportions of phyla (**A**) or genus (**B**), based on the 18S dataset, according to the ‘Time’ variable.

**Figure 4 jof-10-00428-f004:**
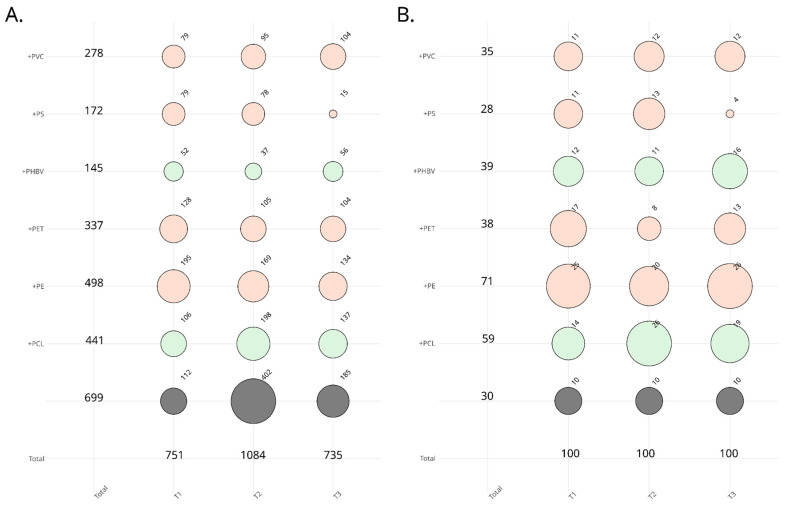
Bubble plot showing the proportions of isolates per isolation medium (BH+PVC, +PS, +PHBV, +PET, +PE, +PCL) from the initial collection (**A**), or stratified and weighted sampling (**B**) to select representative isolates from this collection.

**Figure 5 jof-10-00428-f005:**
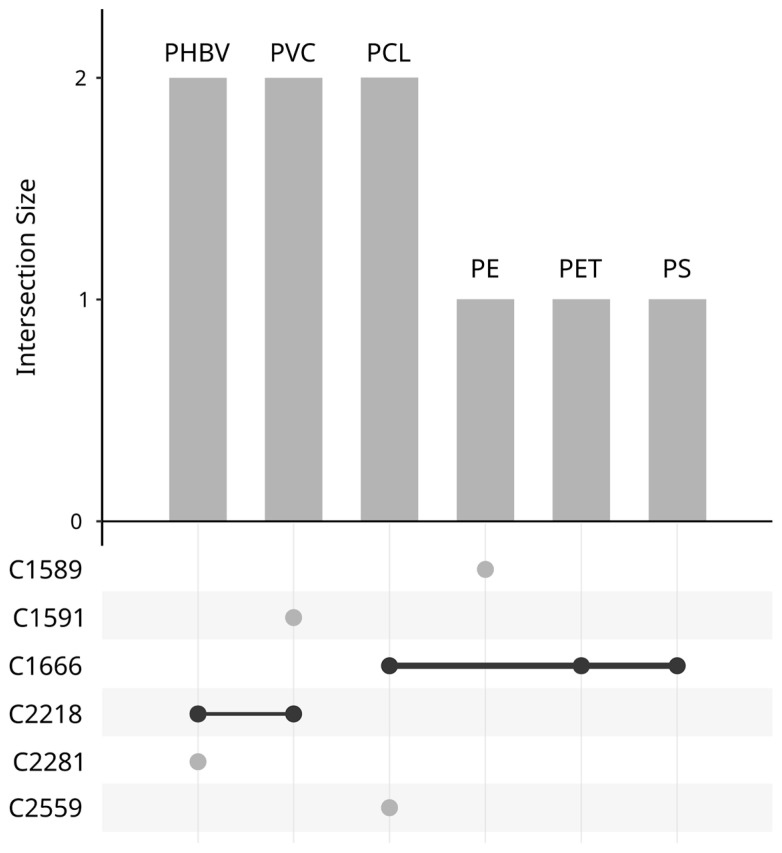
UpSet plot representing the most promising isolates utilizing either a single polymer (light grey circles) or several polymers (dark grey circles) as carbon sources.

## Data Availability

The data produced in this study can be found at https://www.ebi.ac.uk/ena/browser/home (accessed on 30 March 2024, project number PRJNA1081862, release date March 2024).

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
