# Peer review of "Colonization and Biodegradation Potential of Fungal Communities on Immersed Polystyrene vs. Biodegradable Plastics: A Time Series Study in a Marina Environment"

_jof, 2024, doi:10.3390/jof10060428_

Round 1

Reviewer 1 Report

This paper compared the fungal communities colonizing foamed polystyrene and alternative biodegradable plastic with surrounding seawater in marine environments using amplicon sequencing technology. The author also explored the culturable fungi that utilise polymers as carbon sources. While the topic is of interest to general readers, there are several areas in need of improvement.

Key Problems and Recommendations

1)      The introduction fails to adequately address recent research on marine plastic biodegradation, particularly regarding the fungal colonization of marine plastics. Existing studies have explored the marine fungal community on plastic surfaces (including PS) and their ecology (e.g., Kirstein et al., 2018, Lacerda et al., 2020). Several review papers have updated the list of plastic degraders and the potential driving factors of their community composition and function (Shah et al., 2008; Bhardwaj et al., 2013; Kale et al., 2015; Pathak, 2017).

I didn’t see a clear research question in this study, why choose the 6 types of polymers, the definition of material properties and sampling frequency, and what are the hypotheses? I will recommend that the author summarize current knowledge about fungal communities and their ecology in marine environments and their capacity to colonize different plastic materials (such as the sheet-shaped polymer in this study). Establishing hypotheses for fungal ecology at different times and the colonization patterns on different plastics will aid in interpreting results.

2)      The relevance of the culturable fungi isolated in this study to the plastic degradation in the marine environment is unclear.

For example, the culture media used (such as Martin Rose Agar, Sabouraund-Dextrose, Potato Dextrose agar, and Czapeck) contain ingredients like glucose/sucrose and peptone, which may favour the growth of non-marine-relevant fungi (including bacteria). This could be the reason that 2000 fungal isolates have been enriched using the above media, while <90 fungal ASV were detected in the field experiment using the culture-independent amplicon sequencing method.

Besides, the random sampling of colonies and recultured using a minimum medium (BH) plus the sea salts was also conducted in this study.  The author did screen for isolates with wide oxygen adaptation and growth capability. However, it still lacks relevance with their field study, some factors such as the pH, nutrient levels and aerobic conditions of the immersed plastic conditions are also critical for the growth of their fungal colonizers.  Besides, the growth didn't directly correlate to their capacity of plastic biodegradation. 

3)      The study lacks measurements of plastic degradation, such as mass loss, size reduction, or changes in functional groups, to verify fungal functional capacity. Overall, the culture time of 9 weeks in this study is relatively short to detect polymer degradation. Without functional validation, the ecological significance and functional relevance of fungal taxa dynamics remain unclear and lack novelty and knowledge advancement. 

1)      L128-133: Several material characteristics, including biodegradability, ability to absorb organic pollutants, and toxicity, were also considered in this study. Given that the final comparison was made between 3 foamed PS and 3 biodegradable plastics vs marine water, the relevance of these properties to the composition and function of fungal communities should be reviewed, and the research question should be clearly justified.

2)      L165 would be necessary to provide the quality statistics for sequencing data. Given that the fungal diversity is relatively low in this study, it is necessary to understand whether the DNA quality and sequencing depth are sufficient to capture the diversity of the fungal diversity.

3)      Figure 1, Supplementary Figure 1, given no visible differences for both alpha/beta diversity among these fungal communities, the description in the results is difficult to follow. Besides, Axis 1 and Axis 2 altogether explain <20% variation in the system, a poor explanation capacity. Please replot the figure to highlight the key findings clearly. For example, use different colors representing the polymers and different shapes for sampling times to show the trend along the designed factors or their distance from the control seawater.

Reviewer 2 Report

Actually the manuscript presents a thorough solid research and is well written. I did not found any significant imperfections. The only thing I would advise to improve is to describe the differencies between three used polystyrenes to attempt explain the differences of adhesive fungal communitis. The diference between the ability of communities adhering to different polystyrenes and biodegradable plastics to grow on conventional plastics has been statistically confirmed. However  this relationship most probably is just coincidential, because there are no noticable relationship between different plastics structure.

Actually the manuscript presents a thorough solid research and is well written. I did not found any significant imperfections. The only thing I would advise to improve is to describe the differencies between three used polystyrenes to attempt explain the differences of adhesive fungal communitis. The diference between the ability of communities adhering to different polystyrenes and biodegradable plastics to grow on conventional plastics has been statistically confirmed. However  this relationship most probably is just coincidential, because there are no noticable relationship between different plastics structure.

Author Response

#Reviewer 2 :

Major & detail comments

Actually the manuscript presents a thorough solid research and is well written. I did not found any significant imperfections. The only thing I would advise to improve is to describe the differencies between three used polystyrenes to attempt explain the differences of adhesive fungal communitis. The diference between the ability of communities adhering to different polystyrenes and biodegradable plastics to grow on conventional plastics has been statistically confirmed. However  this relationship most probably is just coincidential, because there are no noticable relationship between different plastics structure.
Thanks a lot for the positive feedback. 
Thank you for this very pertinent comment. Unfortunately, we do not have precise data on the exhaustive composition of the polystyrenes, particularly regarding the presence of additives. The polystyrenes were purchased commercially, and due to the rich diversity of polystyrene products and the unique formulation of each producer, differences (even minor) in the structural properties of the products are inevitably associated with distinct chemical additives that could cause the observed variations in adhesion. Delving further into this aspect in the discussion seems risky, which is why we decided to focus on other variables such as toxicity to marine organisms and the transfer of organic pollutants.

Reviewer 3 Report

Some contradictions with the methodology and results, please see my detailed comments.

Abstract

Line 19: Change "Marina" to "Marine."

The authors chose 18S rRNA instead of the ITS region for fungal classification. In my opinion, ITS has a higher resolution for classifying fungal communities than the 18S region.

The authors should highlight the potential fungal members identified in both culture-dependent and independent analyses, especially the significant fungal members involved in plastic degradation.

Introduction

Line 50: The statement "EPS and XPS litter are frequently encountered in marine ecosystems" requires an appropriate citation.

Overall, the introduction is well written.

Materials and Methods

Line 121: What are the total W/V percentages of the different plastic materials used in the study?

Why were positive and negative controls for plastics not included in the immersion experiments?

How do you confirm that total DNA was extracted from the surface of the plastic? How did you process the plastic materials prior to DNA extraction other than just rinsing?

The PR2 V5.0.1 database is more effective for protist classification than fungal classification, as it emphasizes protists over fungi. What was the reason for selecting 18S instead of ITS, considering that UNITE has more classified sequences for fungi?

How did you optimize the W/V of 10g/L of plastic materials? This amount seems too high for microbial enrichment.

For the culture-dependent approach, why were 18S primers not used? contradicting

Results

Considering only 97 ASVs out of 57 samples seems low. This might be due to the selection of variable regions for fungal classification.

The resolution of Fig. 1 needs improvement.

The captions for Fig. 2 are confusing. It appears that A and C are polymers at the genus level, while B and D are types of plastic at the genus level.

It is interesting that Cladosporium was not shown as a promising isolate in plastic degradation, while Penicillium ASV was only identified in PLA in culture-independent studies. Given that many fungal members produce spores, it is unconvincing that some strains might be air contaminants.

Discussion

Genus names should be italicized throughout the draft; some places are inconsistent.

Line 737: Penicillium was isolated from PBAT and PHBV samples; however, no ASVs were identified in metabarcoding results. Please explain.

Why was there no study on FTIR and SEM analysis? These are important for confirming plastic degradation and functional changes during microbial growth.

Author Response

#Reviewer 3 :

Major comments

Some contradictions with the methodology and results, please see my detailed comments.

Detail comments

Abstract

Line 19: Change "Marina" to "Marine."

With all due respect to Reviewer 3, the term "marina" refers to a complex that includes a leisure port. We wish to retain "marina" as it represents a specific coastal ecosystem.

The authors chose 18S rRNA instead of the ITS region for fungal classification. In my opinion, ITS has a higher resolution for classifying fungal communities than the 18S region.

In fact, we chose a dual approach by amplifying both the 18S and ITS2 regions, as we did in a previous paper (Philippe et al. 2023). However, the sequencing platform failed to obtain the ITS2 data and could only provide us with the 18S data. Considering the results of our previous paper (Philippe et al. 2023), which investigated the colonization of plastic matrices by marine fungi using a metabarcoding approach and showed similar trends with both 18S and ITS2 regions, and considering the publication by Lacerda et al. 2020, which studied fungal communities associated with marine plastic debris using metabarcoding targeting the ITS2 and 18S regions and confirmed the value of using two genetic markers to reveal greater diversity but emphasized that the 18S marker allowed better resolution and the detection of basal fungal lineages not detected by the ITS2 marker, and considering that of the 11 publications focusing on fungal communities associated with marine plastics using metabarcoding (Oberbeckmann et al. 2016, Debroas et al. 2017, Kettner et al. 2017, De Tender et al. 2018, Kirstein et al. 2018, Kettner et al. 2019, Lacerda et al. 2020, Lacerda et al. 2022, Yang et al. 2022, Philippe et al. 2023, Servulo et al. 2023), 9 used the 18S marker, we can conclude that an approach focused solely on 18S here does not introduce significant bias and allows for better comparison with the literature. A paragraph has been added in the ms, part 4.1:

“A singular genetic marker was employed in this study, warranting a special note. This choice stemmed from technical limitations in sequencing, which precluded the acquisition of ITS data, thus restricting our analysis to 18S data exclusively. Despite the recognized potential of utilizing two markers to unveil greater diversity, as underscored by Lacerda et al. 2020, both 18S and ITS datasets yielded congruent trends in a previous work of the fungal plastisphere (Philippe et al., 2023). Moreover, within the spectrum of 11 publications scrutinizing fungal communities linked to marine plastic debris (Oberbeckmann et al., 2016; Debroas et al., 2017; Kettner et al., 2017; De Tender et al., 2018; Kirstein et al., 2018; Kettner et al., 2019; Lacerda et al., 2020; Lacerda et al., 2022; Yang et al., 2022; Philippe et al., 2023; Servulo et al., 2023), nine relied upon the 18S marker. This alignment facilitates a more robust comparison of our findings with existing literature. Lastly, it is noteworthy that the 18S marker is adept at elucidating basal fungal lineages, a feature highlighted by Lacerda et al. in 2020.”

The authors should highlight the potential fungal members identified in both culture-dependent and independent analyses, especially the significant fungal members involved in plastic degradation.

This information was presented in the discussion section, particularly in section 4.3, for isolates identified as Penicillium and Aspergillus. However, it was absent for Gliomastix. Subsequently, it has been included:

“Unfortunately, our metabarcoding dataset did not reveal any ASV affiliated to the genus Gliomastix, a discrepancy that may be attributed to potential biases such as the low abundance of this taxon on the plastic matrix.

Introduction

Line 50: The statement "EPS and XPS litter are frequently encountered in marine ecosystems" requires an appropriate citation.

The reference for this sentence is actually provided at the end of the subsequent sentence, which is a continuation (reference 8). However, recognizing the ambiguity, we opted to remove this sentence as it is redundant with the following one: “Between 2018 and 2020, in OSPAR countries actively monitoring foamed polystyrenes, including Denmark, the Netherlands, Germany, France, Ireland, and Portugal, EPS and XPS pollution constituted 15% of the total number of plastics and 13% of the overall litter found on beaches [8], raising significant environmental concern.”

Overall, the introduction is well written.

Thank you for your comment. We also believe the introduction effectively addresses recent research on marine fungal diversity associated with plastics and the biodegradation of marine plastics.

Materials and Methods

Line 121: What are the total W/V percentages of the different plastic materials used in the study?

With all due respect to reviewer 3, we find it challenging to understand this comment. Immersed polymers cannot be characterized by a variable such as total M/W%. We would have appreciated having information on the molecular weights, for instance, of the submerged plastics, but unfortunately, we do not have access to such data. However, it is important to note, as indicated in the materials and methods section, that all submerged plastics are of uniform size and thickness, ensuring consistent available surface area for adhesion to avoid introducing bias between materials.

Why were positive and negative controls for plastics not included in the immersion experiments?

It is indeed regrettable that control materials, such as wood or glass, were not included in this experiment, as they could have yielded valuable insights. However, the primary aim of this study was to elucidate the colonization dynamics of fungal communities on polystyrene-type polymers in comparison to biodegradable polymers. Given this objective, the inclusion of materials like wood or glass, while potentially informative, was not necessary for the experimental design and specific research question. Currently, in fungal plastisphere studies, the use of control matrices is not yet standard practice, with only 3 out of 8 studies on plastic immersion incorporating such controls. Nevertheless, we do have the 'seawater' control, which is essential for this type of study.

How do you confirm that total DNA was extracted from the surface of the plastic? How did you process the plastic materials prior to DNA extraction other than just rinsing?

Thanks for pointing this out. We have added complementary information in the Materials section:

Sterile DNA extraction was performed from the 54 sampled plastic materials (1 cm2 piece, extracted using a sterile scalpel blade taking care to only collect the top 2mm of the surface)

The PR2 V5.0.1 database is more effective for protist classification than fungal classification, as it emphasizes protists over fungi. What was the reason for selecting 18S instead of ITS, considering that UNITE has more classified sequences for fungi?

The PR2 database, while widely used for protists, also performs very well for fungi. The corresponding author of this study has contributed to the curation of the fungal section of the PR2 database. Numerous studies using PR2 for 18S metabarcoding data analysis, including the TARA dataset, have produced excellent results in terms of assignation.

In our study, we also performed ASV assignment using the most recent version of SILVA database, but it proved to be less resolutive than PR2, which we preferred (data not shown).

How did you optimize the W/V of 10g/L of plastic materials? This amount seems too high for microbial enrichment.

To determine this value of 10 g/L polymer supplementation, we conducted a literature review that revealed the majority of studies supplement media at 1%. Therefore, we chose 10 g/L as it corresponds to 1% (w/v).

For the culture-dependent approach, why were 18S primers not used? Contradicting

The choice of genetic marker(s) in metabarcoding varies from that in culture. For these particular isolates, which were deemed the most promising, we, as mycologists, did not rely on 18S sequencing for identification. Instead, we determined their affiliation with the genera Penicillium and Aspergillus based on morphological features. Consequently, commonly used markers, such as beta-tubulin for Penicillium and actin for Aspergillus, were utilized for their identification. Nonetheless, to ensure precise identification, we also selected markers known as discriminant for these taxa (ITS and 28S). We believe this is not contradictory; rather, it reflects the divergent approaches employed in metabarcoding versus culture-based methods for identification.

Results

Considering only 97 ASVs out of 57 samples seems low. This might be due to the selection of variable regions for fungal classification.

See previous responses.

The resolution of Fig. 1 needs improvement.

Figure 1 has been optimized

The captions for Fig. 2 are confusing. It appears that A and C are polymers at the genus level, while B and D are types of plastic at the genus level.

Thank you for pointing out this mistake. Indeed, it was not about A and B / C and D but rather A and C and B and D.

It is interesting that Cladosporium was not shown as a promising isolate in plastic degradation, while Penicillium ASV was only identified in PLA in culture-independent studies. Given that many fungal members produce spores, it is unconvincing that some strains might be air contaminants.

Indeed. However, it's crucial to note that while only 6 isolates were characterized as "most promising," this doesn't imply that only 6 isolates are noteworthy among our list of 300 screened ones. Additional isolates show promise, albeit to a lesser degree than the top 6. Undoubtedly, among the screened isolates, some of interest, though not included in the top tier, exhibited a morphology akin to Cladosporium.

Discussion

Genus names should be italicized throughout the draft; some places are inconsistent.

Thanks for this comment. All genus names have been italicized.

Line 737: Penicillium was isolated from PBAT and PHBV samples; however, no ASVs were identified in metabarcoding results. Please explain.

We appreciate your insightful comment, which echoes a previous comment on the disparity in richness between metabarcoding and culture-based methodologies. Additionally, it enabled us to rectify an error concerning DNA extraction in metabarcoding, initially stated as being from 2 cm2, when in fact they were 1 cm2. In response, we have incorporated a paragraph highlighting biases inherent to metabarcoding, including issues with low biomass, DNA extraction challenges with resistant spores, and the heterogeneous distribution of fungi on plastic substrates. Notably, our metabarcoding analyses were conducted on 1 cm2 samples, employing direct DNA extraction, whereas culture-based methods involved cell detachment from larger pieces (specifically, half of a 4 x 3 cm piece, equivalent to 3 cm2). Consequently, one hypothesis to account for the observed differences in richness between these methodologies could be the uneven distribution of fungal microorganisms on plastics, leading to a less straightforward detection in metabarcoding (which analyzes 1 cm2 samples) compared to culture (where cells are detached from 3 cm2 areas).

A paragraph has been added part 4.3 in the discussion:

“A notable observation is the relative inconsistency observed between the identified taxa and the metabarcoding dataset. While we have established congruence between these two methods for Aspergillus fumigatus and Penicillium brevicompactum, as evidenced by ASVs associated with these genera being detected on the same polymers, it is crucial to acknowledge disparities for other Penicillium species and Gliomastix murorum. Various sources of bias are evident, including (i) the challenge of detecting low biomass through metabarcoding, despite the presumed sufficiency of a single spore for growth on culture media, (ii) the complexity of DNA extraction from resistant spores prior to molecular analysis, and (iii) the inherent heterogeneity in fungal distribution on plastic surfaces. Our metabarcoding analyses were conducted on 1 cm2 samples (direct DNA extraction), whereas in culture, cells were detached from larger pieces (equivalent to 3 cm2). Thus, one plausible explanation for the observed differences in species richness between these methodologies could be the uneven distribution of fungal microorganisms on plastics, leading to less straightforward detection in metabarcoding compared to culture.”

Why was there no study on FTIR and SEM analysis? These are important for confirming plastic degradation and functional changes during microbial growth.

This manuscript primarily delves into the diversity and ecological dynamics of fungal communities associated with marine plastic debris. While we acknowledge a “biotech" aspect, as indicated by our screening for polymer-utilizing capabilities, our main focus remains on the broader ecological context. We concur that a comprehensive analysis employing various methodologies (such as FTIR, mass loss, and C13-labeling) on the most promising isolates is now important. Nonetheless, such endeavors will be the subject of subsequent publications, as outlined in the discussion within section 4.2, "Ability to utilize polymers as carbon sources".

“The most promising isolates now require in-depth complementary analyses to validate the biodegradation of polymer(s) through various dedicated methods (e.g. mass weight, FTIR, 13C-labelled polymers, etc.) as it was performed on Alternaria alternata FB1 [64], Zalerion maritima [65], or Rhodotorula mucilaginosa [66] on PE.”

Integrating this supplementary information into the present manuscript would unduly extend its length, thereby compromising clarity and focus amidst the wealth of already existing information.

Reviewer 4 Report

The presented article is dedicated to the serious and interesting problem, namely: participation of marine fungi in colonization and degradation of floating plastic wastes. In whole, the presented data describe taxonomic diversity of the fungi isolated from the plastics and ability of some isolated strains (selected by random choice) to grow on the model plastics.

These data enrich our knowledge in the mentioned area.

Nevertheless, there are yet some questions related with the applied methodical approaches. I recommend to expand discussion on the methods to explain some possible methodical limitations if any. The methods are:1) desorption quantification of the fungi from marine plastics, 2) preparing of the model plastics for lab experiments (similarity of model and marine plastics), and 3) evaluation of the plastic degradation in the experiments. This expanded discussion will be very helpful for the readers. 

1) Lines 211-213: “A cell-detachment procedure involving a 3 × 5 s sonication step at 40 kHz using a sterilized probe was performed on collected plastic pieces, as described by [45], to detach cells from the plastic matrix.” The mentioned work [45] presents the short information, namely: “Briefly, cell detachment pre-treatment was performed using 1 mM pyrophosphate (30 min at room temperature in the dark) followed by a sonication step (3 × 5 s, 40 kHz, 30% amplitude, sterilized probe Branson SLPe).” Meanwhile, various fungi have different adhesive abilities. Thus, this detachment procedure is some kind of selection which results in crop of detached fungal cells with a weaker adhesion (while the most interesting objects for the plastic degradation are species with the strong adhesive abilities). It could be that primary group of the degrading fungi grows through the plastics while the plastic surface only is covered with the secondary fungal habitants. Please discuss this problem here or in the “Discussion” section below.

2) Lines 222-223, 252-253, 283-285 in the methodical section mention supplementation of the inoculated media with the 6 different model polymers. However, the authors forgot to describe processes of their preparing. I guess that it was the same procedure which they have already described in their former article [45]. Nevertheless, authors have to describe preparing of these polymer additives. It would be nice also to show similarity of the modeled plastics with the marine plastic wastes or – if the authors cannot – at least discuss the principal similarity.

3) Lines 310-312: ability of plastic biodegradation was suggested by fungal growth in presence of modeled plastics. The logic is correct: 1) if the fungi grew, then they used a carbon substrate, 2) if they used a carbon substrate, then they utilized plastic. However, it is necessary to estimate the share of consumed plasticÑ‹ under favorable conditions, at least semi-quantitatively. Is it possible to guess what share of plastics is destroyed (0.01% or 1%)? This discussion will be very essential and helpful for readers to evaluate the role of fungi in the marine plastisphere. 

Author Response

#Reviewer 4 :

Major comments

The presented article is dedicated to the serious and interesting problem, namely: participation of marine fungi in colonization and degradation of floating plastic wastes. In whole, the presented data describe taxonomic diversity of the fungi isolated from the plastics and ability of some isolated strains (selected by random choice) to grow on the model plastics.

These data enrich our knowledge in the mentioned area.

Nevertheless, there are yet some questions related with the applied methodical approaches. I recommend to expand discussion on the methods to explain some possible methodical limitations if any. The methods are:1) desorption quantification of the fungi from marine plastics, 2) preparing of the model plastics for lab experiments (similarity of model and marine plastics), and 3) evaluation of the plastic degradation in the experiments. This expanded discussion will be very helpful for the readers.

Thanks a lot for your positive comments.

Detail comments

1) Lines 211-213: “A cell-detachment procedure involving a 3 × 5 s sonication step at 40 kHz using a sterilized probe was performed on collected plastic pieces, as described by [45], to detach cells from the plastic matrix.” The mentioned work [45] presents the short information, namely: “Briefly, cell detachment pre-treatment was performed using 1 mM pyrophosphate (30 min at room temperature in the dark) followed by a sonication step (3 × 5 s, 40 kHz, 30% amplitude, sterilized probe Branson SLPe).” Meanwhile, various fungi have different adhesive abilities. Thus, this detachment procedure is some kind of selection which results in crop of detached fungal cells with a weaker adhesion (while the most interesting objects for the plastic degradation are species with the strong adhesive abilities). It could be that primary group of the degrading fungi grows through the plastics while the plastic surface only is covered with the secondary fungal habitants. Please discuss this problem here or in the “Discussion” section below.

This observation aligns with one question from reviewer 3 concerning the surface colonization of plastics. Our observations during sample collection post-immersion indicate that, even after relatively extended immersion periods (9 weeks), colonization was confined to the surface layer. This was corroborated when 1cm x 1cm squares were excised for metabarcoding analyses using a sterile scalpel, revealing a lack of biofilm penetration into the plastic substrate, which visually resembled the control samples.

In terms of fungal cell adhesion to the plastic matrix, variations are evident, influenced by factors such as the production of surfactants like hydrophobins, known to enhance adhesion to hydrophobic substrates (Sanchez, 2020). However, discussing the potential bias in cell extraction via the protocol employed remains complex and nuanced, particularly as not all isolates underwent screening.

2) Lines 222-223, 252-253, 283-285 in the methodical section mention supplementation of the inoculated media with the 6 different model polymers. However, the authors forgot to describe processes of their preparing. I guess that it was the same procedure which they have already described in their former article [45]. Nevertheless, authors have to describe preparing of these polymer additives. It would be nice also to show similarity of the modeled plastics with the marine plastic wastes or – if the authors cannot – at least discuss the principal similarity.

With all due respect to reviewer 4, the information regarding the preparation of the supplemented polymers in the culture media is presented in detail in section 2.3. Nevertheless, given the complexity of the experimental design, we have reiterated this information in the sections specified by reviewer 4:

“Before being supplemented in the different culture media, plastic polymers underwent disinfection in a 70% ethanol bath for 4 hours with agitation. Subsequently, ethanol was evaporated using a water bath at 80°C for 20 minutes for all plastic polymers, except for PCL, which was treated at 45°C to prevent the aggregation of plastic particles”.

Reviewer 4 raises an important point. Unfortunately, we could not screen the utilization capacities of polymers as carbon sources using the same polymers immersed in the marina of the port of Brest. However, our aim was to conduct a broad screening of conventional polymers (PE, PET, PP, and PS) and biodegradable ones (PBHV and PCL), easily accessible and well-characterized (suppliers: GoodFellow and Powdered PolySciences). This approach standardizes the results and allows the scientific community to replicate the experimental design in a FAIR manner.

We added 2 sentences in section 2.3:

“The polymers supplemented in the culture media differ from those immersed in the marina of the port of Brest. This distinction ensures the reproducibility of results, as the supplemented polymers are sourced from standardized international suppliers.”

3) Lines 310-312: ability of plastic biodegradation was suggested by fungal growth in presence of modeled plastics. The logic is correct: 1) if the fungi grew, then they used a carbon substrate, 2) if they used a carbon substrate, then they utilized plastic. However, it is necessary to estimate the share of consumed plasticы under favorable conditions, at least semi-quantitatively. Is it possible to guess what share of plastics is destroyed (0.01% or 1%)? This discussion will be very essential and helpful for readers to evaluate the role of fungi in the marine plastisphere.

We agree that this information is indeed very important and interesting. Nevertheless, in order to determine the proportion of plastics utilized/degraded, it is necessary to conduct a series of additional tests (FTIR, mass loss, 13C-labeled polymers, etc.) to obtain concrete evidence. The combination of approaches we used was aimed at "simply" identifying promising isolates from a culture collection. Any estimation of degradation rates at this stage would be extremely risky, and we wish to avoid producing an erroneous value that could be cited in various scientific articles. We hope reviewer 4 will understand our caution.

Round 2

Reviewer 1 Report

The author has made great improvement with clarification and discussion in reply to comments from the last review. No more comments

The current figure seemed a low resolution and small fonts, would increase readability to enlarge the fond and improve the quality for publication. 

Reviewer 3 Report

Accept with Minor editing of English language

NA